

# Characteristics of the complete mitochondrial genome of *Suhpalacsa longialata* (Neuroptera, Ascalaphidae) and its phylogenetic implications

Xin-Yan Gao[1],*, Yin-Yin Cai[1],*, Dan-Na Yu[1,2], Kenneth B. Storey[3] and Jia-Yong Zhang[1,2]

[1] College of Chemistry and Life Science, Zhejiang Normal University, Jinhua, Zhejiang, China
[2] Key Lab of Wildlife Biotechnology, Conservation and Utilization of Zhejiang Province, Zhejiang Normal University, Jinhua, Zhejiang, China
[3] Department of Biology, Carleton University, Ottawa, ON, Canada
* These authors contributed equally to this work.

## ABSTRACT

The owlflies (Family Ascalaphidae) belong to the Neuroptera but are often mistaken as dragonflies because of morphological characters. To date, only three mitochondrial genomes of Ascalaphidae, namely *Libelloides macaronius*; *Ascaloptynx appendiculatus*; *Ascalohybris subjacens*, are published in GenBank, meaning that they are greatly under-represented in comparison with the 430 described species reported in this family. In this study, we sequenced and described the complete mitochondrial genome of *Suhpalacsa longialata* (Neuroptera, Ascalaphidae). The total length of the *S. longialata* mitogenome was 15,911 bp, which is the longest known to date among the available family members of Ascalaphidae. However, the size of each gene was similar to the other three Ascalaphidae species. The *S. longialata* mitogenome included a transposition of tRNA$^{Cys}$ and tRNA$^{Trp}$ genes and formed an unusual gene arrangement tRNA$^{Cys}$-tRNA$^{Trp}$-tRNA$^{Tyr}$ (CWY). It is likely that the transposition occurred by a duplication of both genes followed by random loss of partial duplicated genes. The nucleotide composition of the *S. longialata* mitogenome was as follows: $A = 41.0\%$, $T = 33.8\%$, $C = 15.5\%$, $G = 9.7\%$. Both Bayesian inference and ML analyses strongly supported *S. longialata* as a sister clade to (*Ascalohybris subjacens* + *L. macaronius*), and indicated that Ascalaphidae is not monophyletic.

## INTRODUCTION

The study of mitochondrial genomes (mitogenomes) is of great interest to many scientific fields, including molecular evolution and evolutionary genomics (*Avise et al., 1987*; *Salvato et al., 2008*). Insect mitochondrial genomes are usually a double-stranded circular molecule with a length of 14–20 kbp, including 13 protein-coding genes (PCGs), 22 transfer RNAs (tRNAs), two ribosomal RNAs (rRNAs), and a control region (CR; AT-rich region) (*Boore, 1999*). The most widespread gene arrangement in insect mtDNAs is

Corresponding author
Jia-Yong Zhang,
zhangjiayong@zjnu.cn

hypothesized to be ancestral for the entire Class Insecta (*Clary & Wolstenholme, 1985*; *Boore, Lavrov & Brown, 1998*; *Cameron et al., 2006*). However, more and more researchers have found other gene rearrangements in mitogenomes, mostly related to tRNAs or non-coding regions often within a selected family or order or these may even define clades at a variety of taxonomic scales below the ordinal level (*Beard, Hamm & Collins, 1993*; *Mitchell, Cockburn & Seawright, 1993*; *Cameron & Whiting, 2008*; *Salvato et al., 2008*; *McMahon, Hayward & Kathirithamby, 2009*; *Cameron, 2014a*). Consequently, the particular gene arrangement becomes a significant marker to delimit taxonomic boundaries. Furthermore, the mitogenome has been increasingly used to reconstruct phylogenetic relationships because of its simple genetic structure, maternal inheritance, and high evolutionary rate properties (*Boyce, Zwick & Aquadro, 1989*; *Sheffield et al., 2008*; *Jia & Higgs, 2008*; *Du et al., 2017*).

The insect Order Neuroptera contains approximately 6,000 species worldwide (*Aspöck, 2002*; *Haring & Aspöck, 2004*). Known as net-winged insects, adults usually possess functional membranous wings with an extensive network of veins and cross-veins (*Beckenbach & Stewart, 2008*). The fossil record of Neuroptera dates back to the Late Permian and indicates that they were a major group of insect fauna during the early diversification of the Holometabola (*Aspöck, 2002*). Therefore, their phylogenetic position is likely to have had a key influence on the subsequent evolution of insects (*Beckenbach & Stewart, 2008*). To date, only 42 mitochondrial genomes of Neuroptera are available in databases (*Beckenbach & Stewart, 2008*; *Cameron et al., 2009*; *Haruyama et al., 2011*; *Negrisolo, Babbucci & Patarnello, 2011*; *He et al., 2012*; *Zhao et al., 2013*, *2016*; *Wang et al., 2013*; *Cheng et al., 2014*, *2015*; *Yan et al., 2014*; *Zhang & Wang, 2016*; *Lan et al., 2016*; *Zhang & Yang, 2017*; *Song, Lin & Zhao, 2018*) and this includes 21 partial mitochondrial genomes. Hence, there is a great need to add data for more Neuroptera species in order to be able to analyze phylogenetic relationships both within this group and to further understand relationships within the Holometabola.

The owlflies (Family Ascalaphidae) belong to the Neuroptera but are often mistaken as dragonflies because of their morphological similarity. The larvae and adults of Ascalaphidae are usually predaceous and so they play an important role in maintaining ecological balance and pest control if they are well applied. At present, only three mitochondrial genomes of Ascalaphidae, namely *Libelloides macaronius* (Scopoli, 1763) (*Negrisolo, Babbucci & Patarnello, 2011*); *Ascaloptynx appendiculatus* (Fabricius, 1793) (*Beckenbach & Stewart, 2008*); *Ascalohybris subjacens* (Walker, 1853) (*Cheng et al., 2014*), are published in GenBank, meaning that they are greatly under-represented in comparison with the 430 described species reported in this family (*Stange, 2004*). These three published genomes show substantial gene rearrangements in contrast to those of the assumed ancestral insects (*Beckenbach & Stewart, 2008*; *Negrisolo, Babbucci & Patarnello, 2011*; *Cheng et al., 2014*) and it is unclear if the mitogenome of any of these species represents the common condition within the Ascalaphidae. The monophyly of Ascalaphidae was supported by *Wang et al. (2017)* and *Song, Lin & Zhao (2018)*, while the monophyly of Myrmeleontidae did not recovered by *Wang et al. (2017)* because the monophyly of Ascalaphidae clustered into the clade of Myrmeleontidae. Increasing

the number of sequenced species within the Neuroptera will be very helpful for phylogenetic reconstructions of Neuroptera relationships. Hence, in the present study we sequenced the complete mitogenome of *Suhpalacsa longialata* Yang 1992 (Neuroptera, Ascalaphidae) and analyzed its genomic structure and composition in comparison with the other three Ascalaphidae species including determining nucleotide composition, gene order, codon usage, and secondary structure of tRNAs. Additionally, we also analyzed evolutionary relationships within Neuroptera using Megaloptera as outgroups to discuss the relationship between Ascalaphidae and Myrmeleontidae, and the relationships of inter-families of Neuroptera.

## MATERIALS AND METHODS

### Sample origin and DNA extraction

The sample of an adult *S. longialata* used for sequencing was collected from Hangzhou, Zhejiang province, China in July 2017 by LP Zhang. The specimen was identified by JY Zhang and preserved in 100% ethanol at −40 °C in the lab of JY Zhang. Total DNA was isolated from one foreleg of *S. longialata* using Ezup Column Animal Genomic DNA Purification Kit (Sangon Biotech Company, Shanghai, China) according to the manufacturer's protocol.

### PCR amplification and sequencing of *S. longialata* mtDNA

A total of 12 universal primers for polymerase chain reaction (PCR) amplification were modified according to *Simon et al. (2006)*, *Zhang et al. (2008)*, and *Zhang et al. (2018)* (Table S1; Fig. 1) based on the mitogenome sequences of the three-known species of Ascalaphidae (*L. macaronius*, *Ascaloptynx appendiculatus*, and *Ascalohybris subjacens*). Then five specific primers (Table S1; Fig. 1) were designed based on the sequence information from universal primers using Primer Premier 5.0 (PREMIER Biosoft International, CA, USA). All PCR was performed with a BioRADMJMini Personal Thermal Cycler (made in Singapore) using Ta*kara Taq* DNA polymerase (TaKaRa Biotechnology Co., Ltd., Dalian, China) with the following cycling steps: denaturation at 94 °C for 5 min, followed by 35 cycles of 94 °C (50 s for denaturation), 48–60 °C (30–50 s for annealing), and 72 °C (1–3 min elongation), followed by a final elongation at 72 °C for 10 min. PCR reactions were carried out in a 50 µL reaction volume consisting of 32.75 µL sterile deionized water, 5.0 µL 10×PCR buffer (Mg$^{2+}$Free), 5.0 µL MgCl$_2$ (25 mM), 4.0 µL dNTP Mixture (2.5 mM each), 1.0 µL DNA template, 1.0 µL each primer (10 ppm), 0.25 µL Takara Taq DNA polymerase (5 U/µL). All PCR products were visualized by electrophoresis in a 1% agarose gel and sent to Sangon Biotech Company (Shanghai, China) for sequencing of both strands.

### Mitogenome annotation and sequence analyses

The mtDNA sequence was assembled using DNASTAR Package v.6.0 (*Burland, 2000*). The tRNA genes and their cloverleaf secondary structures were determined by MITOS (*Bernt et al., 2013*, available at http://mitos.bioinf.uni-leipzig.de/index.py) using the invertebrate mitogenome genetic code. The CR and rRNA genes were identified by the

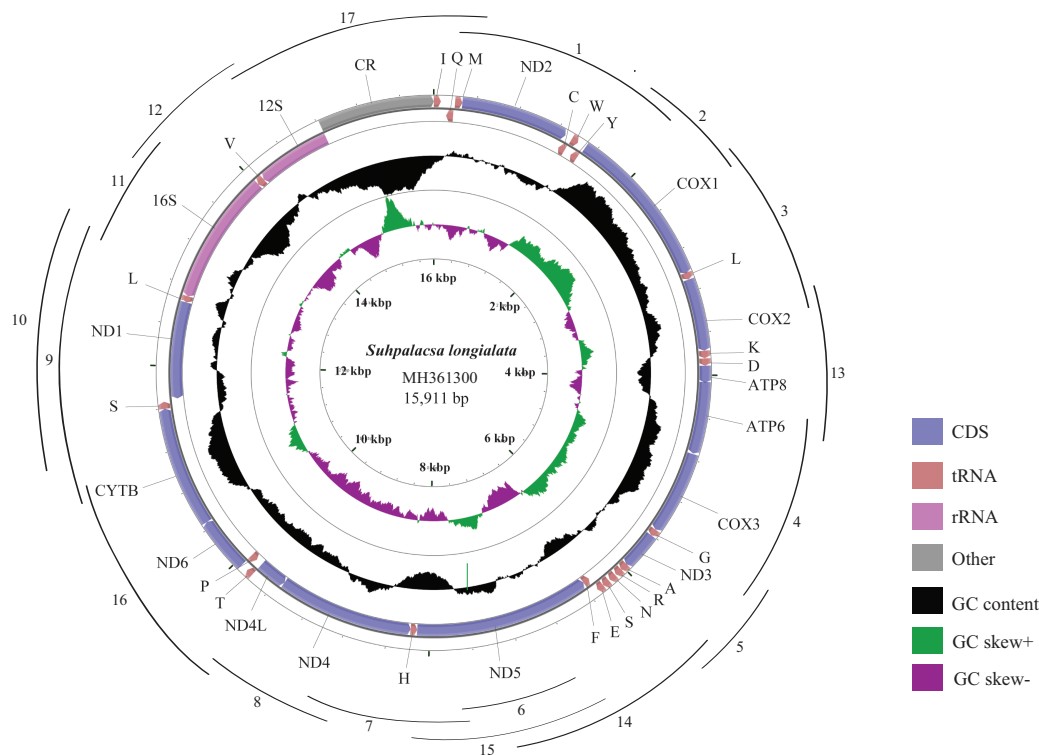

**Figure 1 Mitogenome map of *S. longialata*.** The outermost circle shows the gene map of *S. longialata* and the genes outside the map are coded on the major strand (J-strand), whereas the genes on the inside of the map are coded on the minor strand (N-strand). The middle circle (black) displays the GC content and the paracentral circle (purple & green) displays the GC skew. Both GC content and GC skew are plotted as the deviation from the average value of the total sequence. A total of 17 arcs display the PCR amplification methods. All primers are shown in Table S1.

boundary of tRNA genes (*Thompson et al., 1997*) as well as comparison with homologous sequences of mitogenomes from other species of Ascalaphidae (*Beckenbach & Stewart, 2008*; *Negrisolo, Babbucci & Patarnello, 2011*; *Cheng et al., 2014*). The 13 PCGs were translated to amino acids with the invertebrate mitogenome genetic code and the open reading frames were identified using Mega 7.0 (*Kumar, Stecher & Tamura, 2016*; *Cameron, 2014b*). The nucleotide composition, codon usage, and relative synonymous codon usage (RSCU) were calculated by Mega 7.0 (*Kumar, Stecher & Tamura, 2016*). The GC and AT skews were calculated using the following formulae: AT skew = $(A−T)/(A+T)$, GC skew = $(G−C)/(G+C)$ (*Perna & Kocher, 1995*). A mitogenome map of *S. longialata* was constructed using CG View server V 1.0 (*Grant & Stothard, 2008*).

## Phylogenetic analyses

For Megaloptera as a sister clade to Neuroptera proposed by *Engel, Winterton & Breitkreuz (2018)* and *Peters et al. (2014)*, four species of Megaloptera (*Corydalus cornutus*; *Dysmicohermes ingens*; *Neochauliodes bowringi*; *Sialis hamata*) (*Beckenbach & Stewart, 2008*; *Cameron et al., 2009*; *Li et al., 2015*; *Wang, Liu & Yang, 2016*) were used to as

outgroups in phylogenetic analyses. We downloaded the data from previously sequenced species of Neuroptera as ingroups including *S. longialata* (*Beckenbach & Stewart, 2008*; *Cameron et al., 2009*; *Cheng et al., 2014*, *2015*; *Haruyama et al., 2011*; *He et al., 2012*; *Jiang et al., 2017*; *Lan et al., 2016*; *Negrisolo, Babbucci & Paternello, 2011*; *Wang et al., 2012*, *2013*, *2017*; *Yan et al., 2014*; *Zhao et al., 2013*, *2016*; *Zhang & Wang, 2016*; *Zhang & Yang, 2017*) to discuss family-level phylogenetic relationships of Neuroptera. Accession numbers of all mitochondrial genomes are listed in Table S2. Nucleotide sequences of the 13 PCGs were employed for construction of Bayesian inference (BI) and maximum likelihood (ML) phylogenetic trees according to *Cheng et al. (2016)* and *Zhang et al. (2018)*. DNA alignment was acquired from the amino acid alignment of the 13 PCGs using Clustal W in Mega 7.0 (*Kumar, Stecher & Tamura, 2016*), and the conserved regions were found by Gblock 0.91b (*Castresana, 2000*). We estimated the best partitioning scheme and model by the program PartionFinder 1.1.1 (*Lanfear et al., 2012*) on the basis of Bayesian information criterion. The ML tree was constructed in RAxML 8.2.0 with the best model of GTRGAMMA and the branch support inferred from 1,000 bootstrap replications (*Stamatakis, 2014*). BI analysis was carried out in MrBayes 3.2 (*Ronquist et al., 2012*) with the model of GTR + I + G; the analysis was set for 10 million generations with sampling every 1,000 generations; the initial 25% of generations was discarded as burn-in. Because long branch attraction (LBA) can cause a wrong relationship (*Bergsten, 2005*; *Philippe et al., 2005*), we obtained a second data set using 40 species of Neuroptera (40SN) as the ingroup by excluding *Semidalis aleyrodiformis*, *Coniopteryx* sp., and *Dilar* sp. that showed LBA. The ML and the BI analyses of data 40SN were then performed as above.

## RESULTS AND DISCUSSION

### Mitogenome organization and structure

The complete mitogenome of *S. longialata* is a double-stranded circular DNA molecule with a length of 15,911 bp (Fig. 1) that has been submitted to GenBank under the accession number MH361300. It encodes the entire set of 37 mitochondrial genes including 13 PCGs, 22 tRNA genes, and two rRNA genes that are typically present in metazoan mitogenomes (*Wolstenholme, 1992*). In addition, the gene arrangement of *S. longialata* is similar to the assumed common ancestor of insects (*Mueller & Boore, 2005*; *Yu et al., 2007*; *Erler et al., 2010*; *Li et al., 2011*, *2012a*, *2012b*), with the exception of the tRNA$^{Trp}$-tRNA$^{Cys}$-tRNA$^{Tyr}$ (WCY) triplet. *S. longialata* possessed an unusual gene order of tRNA$^{Cys}$-tRNA$^{Trp}$-tRNA$^{Tyr}$ (CWY) (Fig. 1), which also occurred in the other species of Ascalaphidae available in the GenBank database (*Beckenbach & Stewart, 2008*; *Negrisolo, Babbucci & Paternello, 2011*; *Cheng et al., 2014*). In addition, the transposition of tRNA$^{Cys}$ and tRNA$^{Trp}$ genes has also been found in other families within the Neuroptera, including Dilaridae, Hemerobiidae, Mantispidae, Berothidae, Ithonidae, Chrysopidae, Psychopsidae, Nymphidae, Nemopteridae, and Myrmeleontidae (*Wang et al., 2017*; *Song, Lin & Zhao, 2018*), but not in the other neuropterid orders. Thus, it is widely acknowledged that it may be synapomorphic for the Neuroptera (*Cameron et al., 2009*; *Beckenbach & Stewart, 2008*; *Haruyama et al., 2011*; *Negrisolo, Babbucci & Paternello, 2011*; *He et al., 2012*; *Zhao et al., 2013*; *Yan et al., 2014*).
**Table 1 Base composition of the mitochondrial genomes of four species of *Ascalaphidae*.**

| Region | *S. longialata* | | | | *L. macaronius* | | | | *A. appendiculatus* | | | | *A. subjacens* | | | |
|---|---|---|---|---|---|---|---|---|---|---|---|---|---|---|---|---|
| | Length (bp) | AT% | AT-skew | GC-skew | Length (bp) | AT% | AT-skew | GC-skew | Length (bp) | AT% | AT-skew | GC-skew | Length (bp) | AT% | AT-skew | GC-skew |
| Whole genome | 1,5911 | 74.8 | 0.096 | −0.230 | 15,890 | 74.5 | 0.071 | −0.176 | 15,877 | 75.5 | 0.068 | −0.205 | 15,873 | 75.7 | 0.054 | −0.177 |
| Protein-coding genes | 11,169 | 73.0 | 0.090 | −0.234 | 11,181 | 73.1 | 0.078 | −0.182 | 11,169 | 74.0 | 0.059 | −0.338 | 11,183 | 74.1 | 0.050 | −0.169 |
| Ribosomal RNA genes | 2,053 | 77.8 | 0.159 | −0.297 | 2,095 | 76.4 | 0.094 | −0.241 | 2,078 | 78.6 | 0.125 | −0.280 | 2,094 | 77.8 | 0.108 | −0.270 |
| Transfer RNA genes | 1,476 | 76.2 | 0.055 | −0.122 | 1,471 | 75.6 | 0.037 | −0.115 | 1,464 | 75.5 | 0.057 | −0.135 | 1,466 | 77.7 | 0.037 | −0.135 |
| *A+T*-rich region | 1,088 | 85.1 | 0.086 | −0.168 | 1,049 | 84.5 | 0.030 | 0.006 | 1,066 | 85.7 | 0.048 | −0.077 | 1,051 | 86.2 | 0.035 | −0.014 |

The duplication-random loss model may be a possible explanation for the transposition of contiguous genes. Similar to the report by *Beckenbach & Stewart (2008)*, it is likely that the tRNA$^{Trp}$-tRNA$^{Cys}$ (WC) genes were duplicated in tandem to form a tRNA cluster WCWC, which was then followed by random loss of partial duplicated genes to produce the final CW gene order.

The mitogenome of *S. longialata* (15,911 bp) is the longest as compared with those of other Ascalaphidae species, whose mitogenomes range from 15,873 to 15,890 bp. The greater length of the *S. longialata* mitogenome is due largely to 16 intergenic regions ranging from 1 to 54 bp and a long typical *A+T*-rich region (1,088 bp) as compared to 1,049 bp for *L. macaronius* (*Negrisolo, Babbucci & Patarnello, 2011*), 1,066 bp for *Ascaloptynx appendiculatus* (*Beckenbach & Stewart, 2008*), and 1,051 bp for *Ascalohybris subjacens* (*Cheng et al., 2014*). The nucleotide composition of the *S. longialata* mitogenome is as follows: $A = 41.0\%$, $T = 33.8\%$, $C = 15.5\%$, $G = 9.7\%$. It is obvious that the *S. longialata* had a strong *A+T* bias of 74.8%, which is similar to other species of the Ascalaphidae: 74.5% for *L. macaronius*; 75.5% for *Ascaloptynx appendiculatus*; 75.7% for *Ascalohybris subjacens* (*Beckenbach & Stewart, 2008*; *Negrisolo, Babbucci & Patarnello, 2011*; *Cheng et al., 2014*) (Table 1). The high *A+T* bias was found in PCGs, rRNA genes, tRNA genes, and the CR. Previous studies pointed out that the strand bias in nucleotide composition may be attributed to mutational damage primarily affecting the lagging strand during asymmetric replication (*Francino & Ochman, 1997*; *Hassanin, Leger & Deutsch, 2005*). The skew statistics indicated that *S. longialata* had a positive AT-skew and negative GC-skew (Table 1).

## Protein-coding genes and codon usages

Nine PCGs (ND2, COX1, COX2, ATP8, ATP6, COX3, ND3, ND6, and CYTB) were located on the major strand (J-strand) with the remaining PCGs on the minor strand
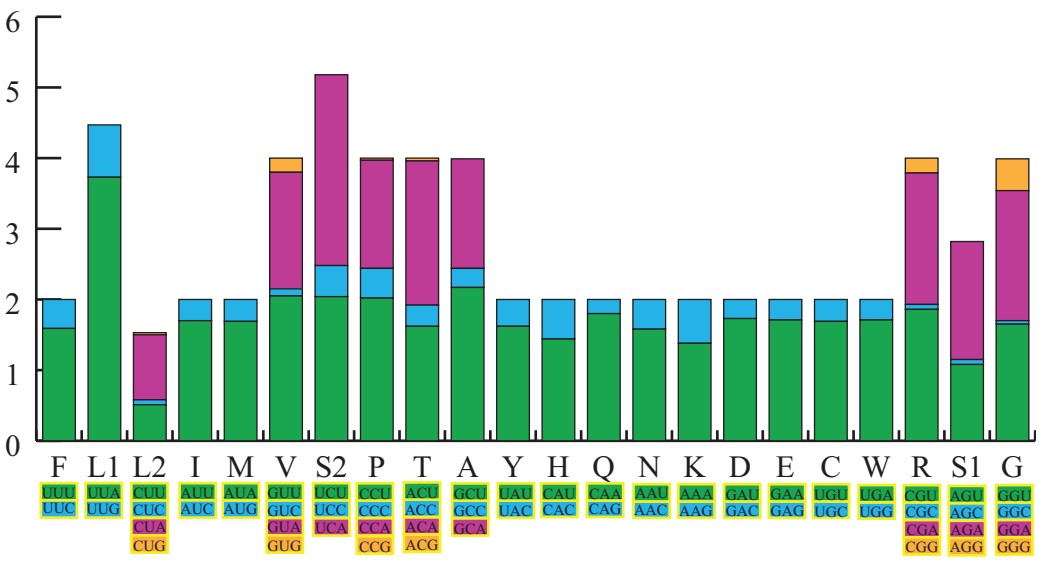

**Figure 2 The relative synonymous codon usage (RSCU) in the *S. longialata* mitogenome.** Codon families are provided on the *x*-axis along with the different combinations of synonymous codons that code for that amino acid. RSCU are provided on the *y*-axis.

(N-strand). All PCGs genes used ATN (*N* represents *A*, *G*, *C*, or *T*) as initiation codons, which have been accepted as the canonical mitochondrial start codons for insect mitogenomes (*Wolstenholme, 1992*). Termination codons for *S. longialata* were mostly complete (TAA) with some incomplete (TA or *T*). Such incomplete stop codons have been found in various insect species (*Ma et al., 2015*; *Nardi et al., 2001*; *Fenn, Cameron & Whiting, 2007*), and it has been determined that incomplete stop codons can produce functional stop codons in polycistronic transcription cleavage and polyadenylation processes (*Ojala, Montoya & Attardi, 1981*). The only exception was detected in ND1, where *S. longialata* exhibited TAG as the stop codon. The infrequent use of TAG may be because of the high *A+T* composition of the PCGs, although TAG is the conservative stop codon in most insect mitogenomes (*Liu et al., 2015*). However, in the other three published Ascalaphidae mitogenomes, COX1 of *L. macaronius* (*Negrisolo, Babbucci & Paternello, 2011*), *Ascaloptynx appendiculatus* (*Beckenbach & Stewart, 2008*), and *Ascalohybris subjacens* (*Cheng et al., 2014*) used ACG as the start codons, and ND1 of *Ascalohybris subjacens* used TTG. The other start/stop codons were identical to the *S. longialata* situation.

The total length of the 13 PCGs in the *S. longialata* mitogenome was 11,169 bp, with an average AT content of 73.0%. The PCGs displayed *A*-skews (*A* > *T*) and *C*-skews (*C* > *G*) (Table 1). We calculated the RSCU of the *S. longialata* mitogenome, excluding stop codons (Fig. 2). The RSCU proved that codons with *A* or *T* in the third position are always overused when compared to the other synonymous codons. The codons of amino acids being NNW (NNA/NNU) were higher than 1.0 without exception in *S. longialata*. The most frequently encoded amino acids were Leu (UUR), Phe, Ile (>300), and the least frequently used amino acid was Cys (<45) (Table S3), which was similar to the other Ascalaphidae mitogenomes (Fig. 2).
## Ribosomal and transfer RNAs

The mtDNA of *S. longialata* contained the entire content of 2 rRNAs and 22 tRNAs genes that were also found in other neuropterid mitogenomes (*Boore, 1999*; *Song, Lin & Zhao, 2018*; *Wang et al., 2017*). The 16S rRNA gene with a length of 1,314 bp was located between tRNA$^{Leu}$ (CUN) and tRNA$^{Val}$ whereas the 12S rRNA gene with a size of 739 bp was located between tRNA$^{Val}$ and the CR; these locations were also detected in the other ascalaphid owlfly species (*Beckenbach & Stewart, 2008*; *Negrisolo, Babbucci & Patarnello, 2011*; *Cheng et al., 2014*). The AT content of rRNAs in the *S. longialata* mitogenome was the highest (77.8%) except for the *A+T*-rich region (85.1%). We found that the AT-skew was strongly positive whereas the GC-skew was highly negative, which showed that the contents of *A* and *C* were higher than those of *T* and *G*, respectively.

The size of the tRNAs was 1,476 bp with an average *A+T* content of 76.2%. Among the 22 tRNAs, most tRNA genes displayed the common cloverleaf secondary structure, whereas the tRNA$^{Ser(AGN)}$ had lost the dihydrouridine (DHU) arm (Fig. 3). The absence of this arm in tRNA$^{Ser(AGN)}$ is a typical feature of many insect mtDNAs (*Wolstenholme, 1992*; *Salvato et al., 2008*; *Sheffield et al., 2008*; *Negrisolo, Babbucci & Patarnello, 2011*; *Yan et al., 2014*; *Du et al., 2017*; *Zhang, Song & Zhou, 2008*), and is usually demonstrated to be functional (*Hanada et al., 2001*; *Stewart & Beckenbach, 2003*). We also found that the tRNA$^{Phe}$ and tRNA$^{Leu}$ (CUN) lack the TψC loops. Furthermore, unmatched U–U base pairs were observed in tRNA$^{Trp}$ (Fig. 3).

In terms of the tRNA gene structures of the other three ascalaphid owlflies, the tRNA$^{Phe}$ in *L. macaronius* and *Ascalohybris subjacens* showed the loss of TψC loops, and the tRNA$^{Ser (AGN)}$ in *Ascalohybris subjacens* lost the DHU loop, whereas the tRNA genes of *Ascaloptynx appendiculatus* displayed the typical cloverleaf secondary structure.

## *A+T*-rich region and intergenic regions

Generally speaking, the *A+T*-rich region was the largest non-coding region, which was located between 12S rRNA and tRNA$^{Ile}$. The *A+T*-rich region of *S. longialata* mtDNA having a length of 1,088 bp was the longest when compared to the other three species of Ascalaphidae, for example, the *L. macaronius* (1,049 bp), *Ascaloptynx appendiculatus* (1,066), and *Ascalohybris subjacens* (1,051 bp). Additionally, the composition of *A+T* was 85.1% in *S. longialata*, which was higher than in *L. macaronius* (84.5%) and lower than *Ascaloptynx appendiculatus* (85.7%) and *Ascalohybris subjacens* (86.2%).

The mitochondrial genomes of most insects are compact (*Boore, 1999*), although large intergenic regions occur in some species. In the *S. longialata* mitogenome the longest intergenic region was a 54 bp insertion between tRNA$^{Ile}$ and tRNA$^{Gln}$. This spacer was also present in *L. macaronius*, *Ascaloptynx appendiculatus*, and *Ascalohybris subjacens* and spanned 55, 42, 54 bp, respectively (*Beckenbach & Stewart, 2008*; *Negrisolo, Babbucci & Patarnello, 2011*; *Cheng et al., 2014*). This intergenic region of the four species also shared a 12 bp long congruent motif A(A/G)TTAA(A/C)TAAAT adjacent to tRNA$^{Gln}$. It has previously been reported that this spacer may diverge quickly among different families of the same order (*Negrisolo, Babbucci & Patarnello, 2011*). Aside from this spacer, gaps between genes ranged from 1 to 18 residues in the *S. longialata* sequence.

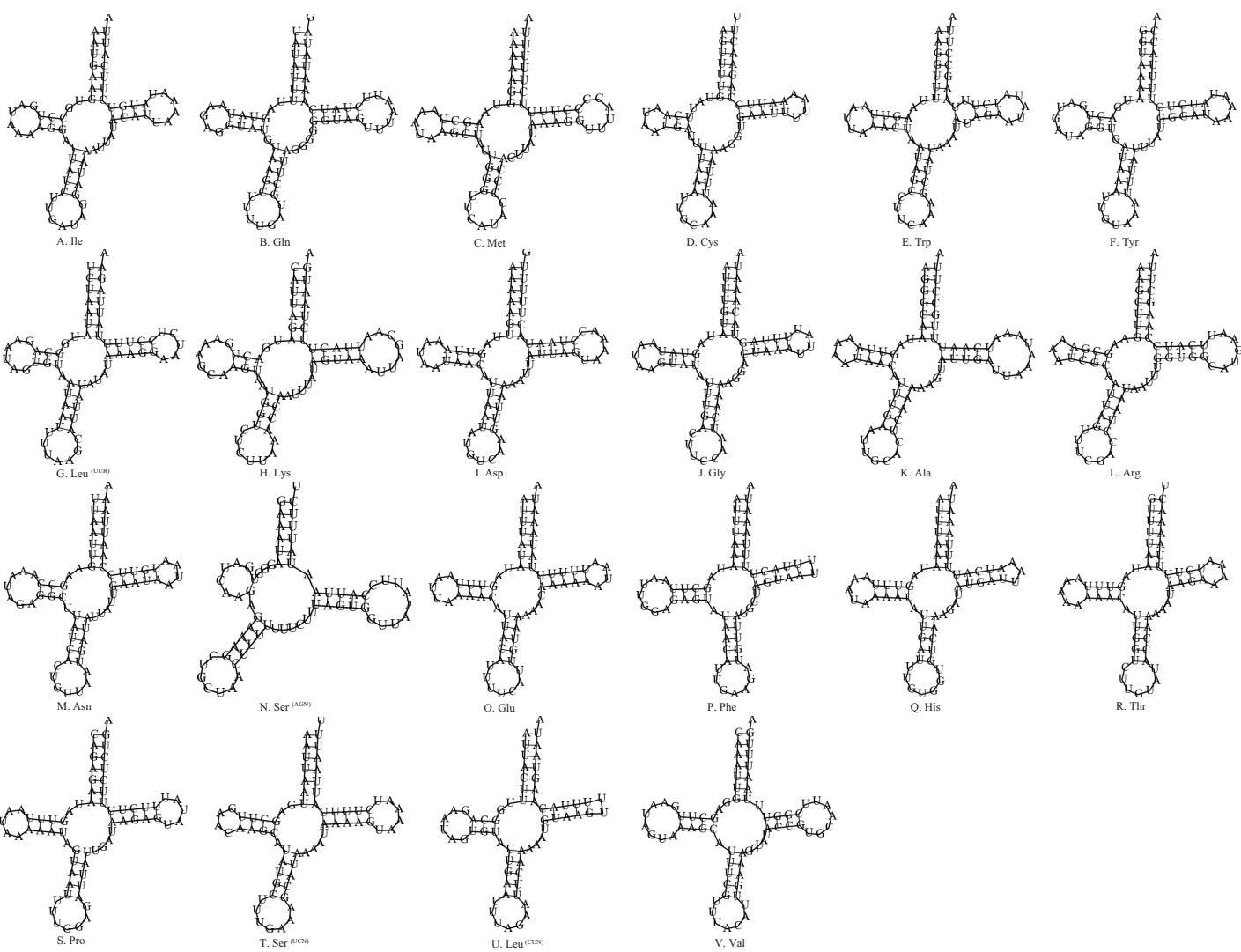

**Figure 3 Secondary structures for 22 transfer RNAs in the *S. longialata* mitogenome.** (A) Ile, (B) Gln, (C) Met, (D) Cys, (E) Trp, (F) Tyr, (G) Leu, (H) Lys, (I) Asp, (J) Gly, (K) Ala, (L) Arg, (M) Asn, (N) Ser, (O) Glu, (P) Phe, (Q) His, (R) Thr, (S) Pro, (T) Ser, (U) Leu, (V) Val.

## Phylogenetic analyses

The phylogenetic relationships including the long-branch attraction species deduced from BI analysis and ML analysis are shown in Fig. 4, and they present somewhat different topologies. In the ML analysis, Hemerobiidae is a sister clade to (Berothidae + Mantispidae) with low support (ML 29). However, in the BI analysis Hemerobiidae is a sister clade to Chrysopidae with high support (BI 1) (Fig. 4). The high support found for both relations (Hemerobiidae + Chrysopidae) and ((Hemerobiidae + Chrysopidae) + (Berothidae + Mantispidae)) only in the BI analysis. In the ML analysis (*Sisyra nigra* + *Climacia areolaris*) is a clade sister to (*Nevrorthus apatelios* + *Nipponeurorthus fuscinervis*), but in BI (*Sisyra nigra* + *Climacia areolaris*) is a clade sister to (*Coniopteryx* sp. + *Semidalis aleyrodiformis*). It has been demonstrated that the LBA artefact will affect both ML and BI tree reconstruction methods (*Huelsenbeck & Hillis, 1993*; *Huelsenbeck, 1995*;

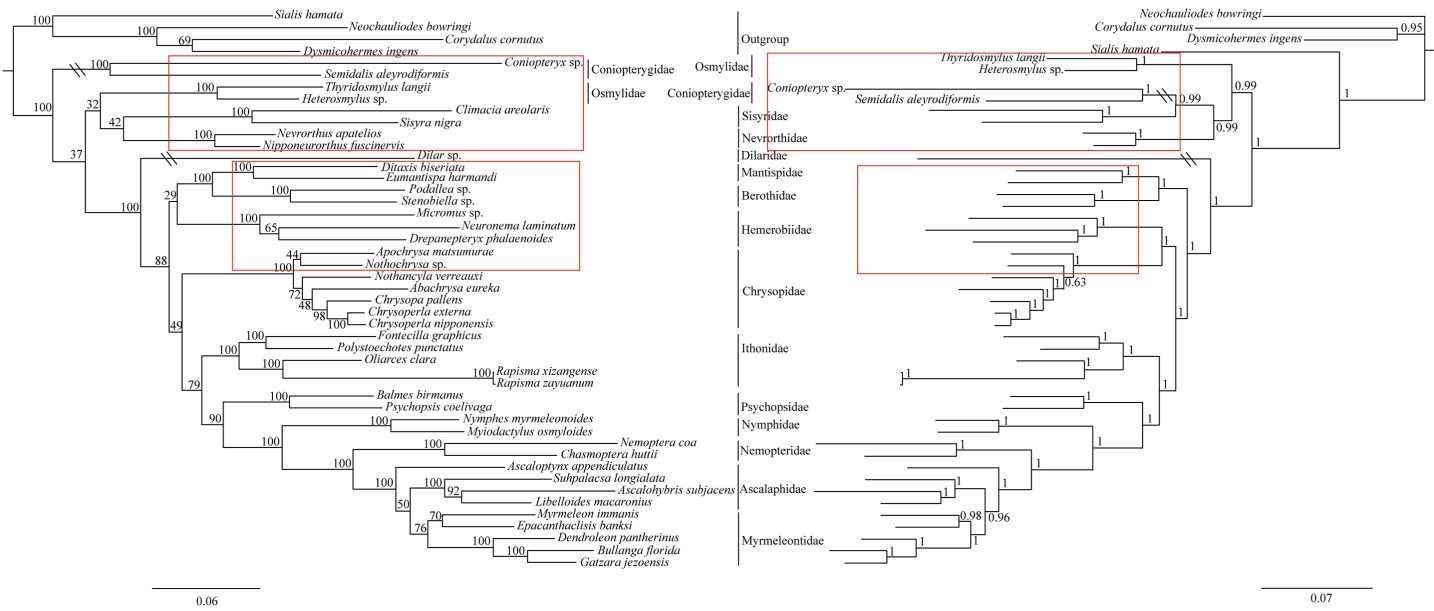

**Figure 4 Phylogenetic relationships of Neuroptera in ML and BI analyses.** The data is includes 43 species of Neuroptera as the ingroup and four species of Megaloptera as the outgroup. The red boxes on the figure mean different topology.

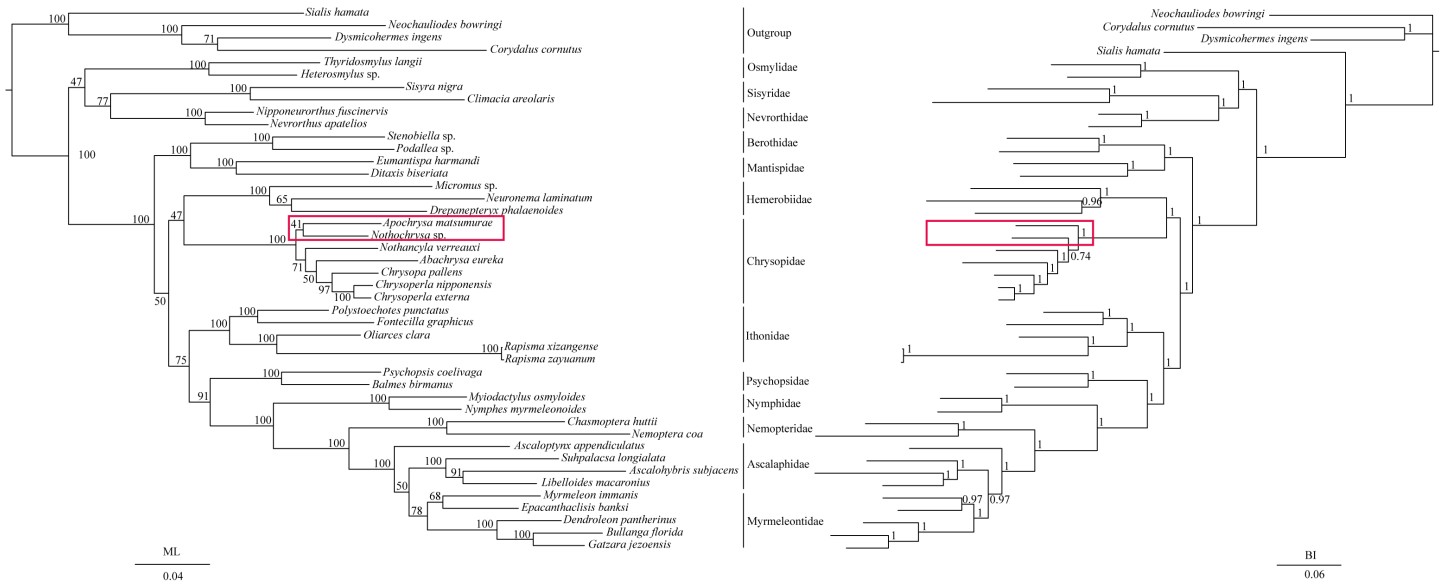

**Figure 5 Phylogenetic relationships of Neuroptera in ML and BI analyses after the elimination of three species (*Semidalis aleyrodiformis, Coniopteryx* sp., *Dilar* sp.).** The data include 40 species of Neuroptera as the ingroup and four species of Megaloptera as the outgroup. The red boxes on the figure mean different topology.

*Philippe, 2000*; *Philippe et al., 2005*). Thus, we propose that the difference between the ML and BI analyses were caused mainly by LBA of *Coniopteryx* sp., *Dilar* sp., and *Semidalis aleyrodiformis*. According to *Bergsten (2005)*, a method excluding long branch taxa can avoid LBA. So we removed three species (*Semidalis aleyrodiformis, Coniopteryx* sp., *Dilar* sp.) and reconstructed the phylogeny of Neuroptera (Fig. 5). In this situation, taking

no account of the outgroup, both the ML and BI phylogenetic trees showed identical topologies and high support values for most clades, except for the internal relations within the family Chrysopidae. *Apochrysa matsumurae* is a sister clade to *Nothochrysa* sp. and then (*Apochrysa matsumurae* + *Nothochrysa* sp.) is a sister clade of (*Nothancyla verreauxi* + (*Abachrysa eureka* + (*Chrysopa pallens* + (*Chrysoperla nipponensis* + *Chrysoperla externa*)))) in ML analysis, whereas in BI analysis (*Apochrysa matsumurae* + (*Nothochrysa* sp. + (*Nothancyla verreauxi* + (*Abachrysa eureka* + (*Chrysopa pallens* + (*Chrysoperla nipponensis* + *Chrysoperla externa*)))))) (Fig. 5). On the whole, this analysis recovers the monophyly of all Neuroptera families except the Ascalaphidae, previously reported as monophyletic by *Wang et al. (2017)* and *Song, Lin & Zhao (2018)*. Two clades of Neuroptera are recovered: one clade is (Osmylidae + (Sisyridae + Nevrorthidae)) and the other clade is (Berothidae + Mantispidae) + ((Hemerobiidae + Chrysopidae) + (Ithonidae + ((Psychopsidae + (Nymphidae + ((Nemopteridae + (*Ascaloptynx appendiculatus* of Ascalaphidae + (Ascalaphidae + Myrmeleontidae))))))))). In the ML analysis LBA existed with all families of Neuroptera (Fig. 4) and Coniopterygidae is recovered as sister clade to the remaining extant Neuroptera, which is consistent with the conclusions of *Wang et al. (2017)* and *Winterton, Hardy & Wiegmann (2010)*, *Winterton et al. (2018)*. By contrast, in the BI analysis (Fig. 4) Osmylidae is recovered as sister clade to (Coniopterygidae + (Sisyridae + Nevrorthidae)). These difference may be caused by the model selection. In this study, we also found that the clade of (Nevrorthidae + Sisyridae) is sister clade of Osmylidae and the clade of ((Nevrorthidae + Sisyridae) + Osmylidae) is sister clade of other extant Neuroptera, excluding Coniopterygidae (Fig. 5), which was also found by *Wang et al. (2017)* and *Winterton, Hardy & Wiegmann (2010)*. The sister relationship of Myrmeleontidae and Ascalaphidae is supported by *Song, Lin & Zhao (2018)*. Myrmeleontidae is monophyletic and Ascalaphidae is not monophyletic in this study. In addition, we make further discussions on the unclear relationship between/within Myrmeleontidae and Ascalaphidae, which were previously controversial since the recent results of mitogenomic phylogeny do not support the monophyly of Myrmeleontidae or Ascalaphidae. (*Yan et al., 2014*; *Lan et al., 2016*; *Winterton et al., 2018*; *Zhao, Zhang & Zhang, 2017*). In this study, the topology is as follows: ((*Myrmeleon immanis* + *Epacanthaclisis banksi*) + (*Dendroleon pantherinus* + (*Bullanga florida* + *Gatzara jezoensis*))) (ML 78, BI 1) (Fig. 5), which supports the monophyly of Myrmeleontidae. Among them, the *S. longialata* that we sequenced is a sister clade to (*Ascalohybris subjacens* + *L. macaronius*), which showed high support both in ML and BI analysis. Because of the increase in species of Neuroptera included in the present analysis, the topologies of the phylogenetic relationships were somewhat different to those of *Wang et al. (2017)* who reported that *Myrmeleon immanis* is a sister clade to (*Dendroleon pantherinus* + (*Ascaloptynx appendiculatus* + (*L. macaronius* + *Ascalohybris subjacens*))). However in present study showed the topology as follows: (*Ascaloptynx appendiculatus* +((*S. longialata* + (*Ascalohybris subjacens* + *L. macaronius*)) + the clade Myrmeleontidae)). We found with the inclusion of *S. longialata* that the monophyly of Ascalaphidae was recovered by *Wang et al. (2017)* and *Song, Lin & Zhao (2018)* did not recover in our results. The monophyly of Ascalaphidae and Myrmeleontidae will need

more species to be added before they can be discussed further. Consequently, we believe that increasing the abundance of mitochondrial genomes of Neuroptera will make a significant difference to resolving and reconstructing the phylogenetic relationships within Neuroptera.

## CONCLUSION

We successfully sequenced the entire mitochondrial genome of *S. longialata*, which showed similar gene characteristics to the other three species of Ascalaphidae. Both BI and ML analyses supported *S. longialata* as a clade sister to (*Ascalohybris subjacens* + *L. macaronius*), but Ascalaphidae is not monophyletic. From the results obtained in the present study, we believe the different topologies of phylogenetic relationships were caused mainly by LBA of *Coniopteryx* sp., *Dilar* sp., and *Semidalis aleyrodiformis*.

## ACKNOWLEDGEMENT

We thank Le-Ping Zhang for help in sample collection.

### Funding

This research was supported by the Zhejiang provincial Natural Science Foundation (Y18C040006) and the National Natural Science Foundation of China (31370042). The funders had no role in study design, data collection and analysis, decision to publish, or preparation of the manuscript.

### Grant Disclosures

The following grant information was disclosed by the authors:
Zhejiang provincial Natural Science Foundation: Y18C040006.
National Natural Science Foundation of China: 31370042.

### Competing Interests

Kenneth B. Storey is an Academic Editor for PeerJ.

### Author Contributions

- Xin-Yan Gao performed the experiments, analyzed the data, prepared figures and/or tables, authored or reviewed drafts of the paper.
- Yin-Yin Cai performed the experiments, analyzed the data, prepared figures and/or tables, authored or reviewed drafts of the paper.
- Dan-Na Yu conceived and designed the experiments, analyzed the data, prepared figures and/or tables, authored or reviewed drafts of the paper.
- Kenneth B. Storey prepared figures and/or tables, authored or reviewed drafts of the paper.
- Jia-Yong Zhang conceived and designed the experiments, analyzed the data, contributed reagents/materials/analysis tools, prepared figures and/or tables, authored or reviewed drafts of the paper, approved the final draft.

## Data Availability

GenBank number: MH361300.

## Supplemental Information

Supplemental information for this article can be found online at http://dx.doi.org/10.7717/peerj.5914#supplemental-information.

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
