# Peer review of "Characteristics of the complete mitochondrial genome of Suhpalacsa longialata (Neuroptera, Ascalaphidae) and its phylogenetic implications"

_PeerJ, doi:10.7717/peerj.5914_

## Round 0.1 · original submission · Major Revisions

Dear colleagues,

Thank you for the submission, this is a very interesting contribution. Both reviewers have made thorough comments, which I suggest be addressed in their entirety by the authors, especially:

1. There are many incomplete and/or ambiguous sentences along the manuscript, as well as some that are conceptually inadequate (eg. the treatment of "basal" lineages as synonymous to "primitive"; or the indication that mtDNA is "superior" to other sources of data). Reviewer #1 did a good job in listing and discussing these, with detail and reference to the literature. So I suggest the authors pay close attention and address these issues.

2. There are comments by both reviewers #1 and #2 that indicate insuficient (or perhaps just unclear) discussion of your phylogenetic results in comparison to previous results. I suggest the authors carefully re-write some of these sections seeking more clarity, and more objective comparison with the literature.

3. Much like reviewer #1, I also feel that the Material and Methods section would benefit from more detailed explanations of what was done. Thankfully, at PeerJ there are no word count limitations and navigation facilitates skipping the methods for those who are not interested. Those who are interested in reproducing your methods will be thankfull for a a more detailed methodology.

Finally, I'd like to clarify that I am classifying this as "Major Revisions" only because there are many small details to be addressed, but each of them are in fact quite minor. As such, I am confident that the authors will be able to address all the comments made by reviewers.

Reviewer 1 ·

Basic reporting

The study has sequenced and examined the mitochondrial genome of the owlfly Suhpalacsa longialata Yang, 1992 (Neuroptera, Ascalaphidae). The genomic structure, organization and composition was compared with published mitochondrial genomes of other species from the same family. Additionally, the authors performed a phylogenetic analysis within Neuroptera, based on 13 protein-coding genes of mitochondria. They used 43 Neuroptera species and an outgroup of four Megaloptera species. In general, the manuscript is well written. Yet, some parts of the text can be reformulated and better explored. I also believe that there are some misinterpretations that need to be rethinking. The figures also need to be improved. Some of the figures and tables could be combined or moved to Supplementary Materials without significantly impacting the manuscript. Please, see detailed comments below.

Experimental design

The research is original and adds to the growing body of knowledge about the mitochondrial genome of insects, as well on the evolutionary relationships within Neuroptera (Insecta). The description of the methods needs to be improved with more detail and clarity. Yet, the method used is in accordance with the objective of the study.

Validity of the findings

The manuscript is very straightforward, and the authors make good use of their data to address questions. However, the manuscript needs to be improved in some aspects, especially with regard to the interpretation of phylogenies. Some of the decisions taken need to be better justified.

Additional comments

My detailed comments are listed below by section, and line number when necessary.

1. TITLE: All the comments regarding the title are just suggestions. First of all, I will suggest adding the name of the order and the family in the title, after the name of the species: “Suhpalacsa longialata (Neuroptera, Ascalaphidae)”. This will make it easier for people to recognize the group to which the species belongs. A second suggestion is to change the reference to phylogeny in the title. As it stands (“… and its phylogeny”), it may imply that it will be a phylogeny only of the species S. longialata, when in fact it is a phylogeny of Neuroptera. My suggestion is something like: “… and its phylogenetic implications”, “… and a phylogeny of Neuroptera”, or “… and with mitochondrial genome phylogeny of Neuroptera”.

2. ABSTRACT: It can be improved to present all sections of the manuscript: background, objectives, results and discussion, and conclusion. PeerJ allows an abstract with no more than approx. 500 words. You have only approx. 95 words. It is important to write a complete but concise abstract to entice potential readers into obtaining a copy of the full paper.

3. INTRODUCTION:
3.1. Species names and the citation of authors: According to the International Code of Zoological Nomenclature, the citation of the author's name of any species is optional (Article 51.1). The norms of PeerJ also do not have instruction on the citation of the author. In this sense, you are free to cite or not. However, I recommend that you cite the original author and date for each species mentioned in the text, from the introduction onwards. You just need to cite the author at the first time the species name appears in the text. This can be important in distinguishing between homonyms and in identifying species-group names.
3.2. Line 53 to 55: I do not understand the reason for relating the word “widespread” to the fact that the pattern of the gene order is found in Drosophila yakuba. I suggest that you remove the reference to Drosophila yakuba. Although not wrong, arrangement is more commonly used than “order”. You could write without losing the meaning: “The most widespread gene arrangement in insect mtDNAs is hypothesized to be ancestral for the entire class Insecta”
3.3. Line 55 to 56: Actually, rearrangements in insects can define clades at a variety of taxonomic scales below the ordinal level. I believe you can make this clear in this sentence, since it is not just for families or orders. I suggest you look at the following article - Cameron, Stephen L. "Insect mitochondrial genomics: implications for evolution and phylogeny." Annual review of entomology 59 (2014): 95-117.
3.4. Line 58 to 59: The statement “mtDNAs are likely to be superior to nuclear DNAs to reconstruct phylogenetic relationships...” is a mistake. Undoubtedly, mtDNAs has many uses, as being important markers for phylogenetics. However, it is neither better nor worse than other sources of data, such as nuclear DNA, morphology and so on. It is a data source like others, with problems and qualities, and that needed to be used and analyzed according to what you want to answer, as well the approach at hand. I suggest you improve this sentence.
3.5. Line 79 to 81: Please, be careful about using the term “basal” or “primitive”. Surely, it is possible to infer the characteristics of extinct ancestral species or genes. However, it is misleading to speak of any species as ‘‘older’’ just based on a tree. I suggest you rethink the use of “basal” and “primitive”, as well rewrite this sentence. Please, see the comment 5.6 for more suggestions on this issue.
3.6. Line 83: At first glance the sentence seems to be incomplete (“Hence, the present study has taken a first step to”). A first step to what? Is the sentence incomplete or left this way intentionally?

4. MATERIALS AND METHODS
4.1. Sample origin and DNA extraction: I missed some additional information. How the sample was collected? Was the sample preserved in ethanol or was it fresh? Was it stored in a freezer? Was the specimen placed in any scientific collection? Are there any considerations about species identification? Is there any relevant information about removal of the leg from the specimen body? Have you made any modifications to the extraction protocol or was it done according to the manufacturer's protocol?
4.2. PCR amplification and sequencing of S. longialata mtDNA: I suggest that you improve the description at lines 95-97 to be reproducible. It is not clear to me how exactly the primers were used in the amplifications. It needs more detail. Did you do exactly as Zhang et al. (2008) did with the amplifications? If yes, it's not entirely clear. It is also unclear how and for what universal, conserved and specific primers were used. Is there any difference between the universal primers and the conserved primers?
4.3. Line 97: Please, cite the manufacturer/developer of Primer Premier 5.0.
4.4. Line 105: The citation of MITOS is not correct. To cite MITOS: M. Bernt, A. Donath, F. Jühling, F. Externbrink, C. Florentz, G. Fritzsch, J. Pütz, M. Middendorf, P. F. Stadler MITOS: Improved de novo Metazoan Mitochondrial Genome Annotation Molecular Phylogenetics and Evolution 2013, 69(2):313-319.
4.5. Line 127: “Consequently, we discovered 9 subsets”. I believe that this sentence is a result obtained from the analysis with PartionFinder 1.1.1. Because of that, it should be described in the results section and not in the methods.
4.6. Line 131 to 133: About the long branch attraction, I also believe that this sentence is a result. It should be described in the results section.

5. RESULTS AND DISCUSSION
5.1. Line 143 to 144: Please, be careful about using the term “basal” or “primitive”. Please, see the comment 3.5 and 5.6 for more suggestions on this issue. I suggest you change “basal ancestor” to “common ancestor”, or just “ancestor”.
5.2. Line 153 to 156: The sentence about the transposition of contiguous genes is confusing. As it is written, it seems that you have empirical evidence to show that occurred a duplication followed by partial loss of genes. However, I belive that you are only suggesting a possible explanation for the pattern found. Beckenbach et al. (2008) just suggested a possible explanation for the reported rearrangement. I suggest that you improve this sentence to make it clear.
5.3. Line 172: This sentence should be in the material and methods, not in the results. Please, remove from the results and add to the material and methods.
5.4. Line 241 to 245: How did you infer that difference were caused mainly by long branch attraction? How did you come to the conclusion that the three species were unstable? I suggest you improve your discussion and provide justification for the long branch affirmation and the decision to remove all three species.
5.5. Line 256 to 258: Describe the differences and similarities between the results of your phylogeny and that recovered by Wang et al. (2017).
5.6. Line 258 to 259: There are two sentences in these lines, and I believe that each one contains misconceptions. You can not say that “Osmylidae is more primitive than Sisyridae” based on your phylogeny. This is a widespread misconception about what phylogenies can tell. A misinterpretation of phylogenies can be strongly influenced by many factors, which I will not discuss completely here. However, a misinterpretation of trees can occur when the species-poor sister group is thought of as “basal” or “early diverging” with respect to its species-rich sister. In this case, the species-poor sister group are mistakenly assumed to be “primitive” and to represent traits of the common ancestor. Since every branching in a tree is rotatable, if you change the position of the terminals just by rotating the nodes, which terminal would you consider as “primitive”? The term primitive is also problematic and should be avoided. It is misleading to speak of any species as ‘‘older’’ just based on a tree, especially without any information about time. If both lineages survive to the present, they continue to evolve at some rate at least for some characteristics throughout evolutionary history. I recommend three articles to read on the topic (listed below). These articles can help you interpret your phylogeny and rewrite this part of the discussion. I believe that there is also a misinterpretation in the sentence “Ascalaphidae branched earlier than the Myrmeleontidae”. According to the results, Ascalaphidae is not monophyletic in relation to Myrmeleontidae. In this sense, “Myrmeleontidae is inside Ascalaphidae”.
Recommendation of articles
Krell, Frank‐T., and Peter S. Cranston. "Which side of the tree is more basal?." Systematic Entomology 29.3 (2004): 279-281.
Crisp, Michael D., and Lyn G. Cook. "Do early branching lineages signify ancestral traits?." Trends in Ecology & Evolution 20.3 (2005): 122-128.
Omland, Kevin E., Lyn G. Cook, and Michael D. Crisp. "Tree thinking for all biology: the problem with reading phylogenies as ladders of progress." BioEssays 30.9 (2008): 854-867.

6. CONCLUSION
6.1. Line 266 to 267: I suggest you rewrite this sentence. You performed the analysis and you got a result. According to the results (the relationships recovered by the phylogenetic analysis), Ascalaphidae is not monophyletic.

7. TABLES
7.1. I would suggest rethinking the captions of all tables. The captions of the tables need to be well explained.
7.2. Table 1: I suggest moving this table to the supplementary material. Although it is very important for other researchers to be able to repeat the work, I believe it is not relevant to be along with the text
7.3. Table 2: You can consider moving this table to the supplementary material as well.
7.4. Table 3: I consider it important to have this table along with the text. However, the dimensions of the table can be better adjusted
7.5. Table 4: I believe that maybe this table is not needed along with the text. Perhaps the table can be moved to supplementary material. You barely use the information in this table for discussion in the text. Also, the table is too large. You need to write what is the abbreviation RSCU.

8. FIGURES
8.1. Figure 1: This is one of the most important figures of the work, however it needs to be improved. With the exception of color legend, all text outside the representation of the mitochondrial genome can be removed (Accession: unknown; Length: 15, 911 bp; and Suhpalacsa longialata complete genome). Color legend is too small, it is not possible to read what is written.
8.2. Figure 2: I believe the description given in the text is probably enough, so it’s not necessary to include this figure.
8.3. Figure 3: The caption of the figure needs to be better explained. What is the Y-axis? What is the X-axis?
8.4. Figure 4: the dimensions of the figure can be better adjusted. Perhaps the figure can be moved to supplementary material.
8.5. Figures 5 to 8: Maybe they can be combined. A figure with ML analysis and another figure with BI analysis. Or, a figure with ML and BI analysis of all species, and another figure with ML and BI analysis without the three “unstable species”.

Reviewer 2 ·

Basic reporting

I think the work is very interesting, the Suhpalacsa longialata species is part of an under-represented family (Ascalaphidae) and part of an important Insecta order with unsolved phylogenetic questions (Neuroptera). In general, the manuscript is clearly written, the figures and tables are adequate. Only the Figure 6 is smaller than the other phylogenetic trees, so it is difficult to read the species names. In Table 4 caption lacks the initials for relative synonymous codon usage (RSCU) and in the Table 2 is a little difficult to understand to which family belong each species. I suggest put the name of the family aligned to the first species name for each group.
Lines 48-51: Lack reference about the conserved order of the genes in insect mitochondrial genomes.
Line 252: Lan et al. 2014 is not in the references, only Lan et al. 2016.

That being said, I would like to point some things that are not clear for me and give some suggestions that I think would improve the work:

The ideas in the first paragraph of the Introduction seem contradictory, it seems genes rearrangements are the main reason for the use of the mitochondrial molecule to solve phylogenetic questions, although the mitochondrial gene order is extremely conserved among the insects.
I think it is important to mention that most of rearrangements in the mitochondrial molecule are related to tRNAs or non-coding regions, and there are few cases that the morphology of the genome represents a synapomorphy, that is the case for the Neuroptera order, but not necessarily for the families in this group. Usually genomes rearrangements are not used as phylogenetic markers (Cameron et al 2006). The main usage of the mitochondrial molecule to solve phylogenetic questions is the use of the sequence to reconstruct the phylogenies and that was the approach used to answer the questions in this work. So, I think it could be better explained why the mitochondrial molecule is good as phylogenetic marker.
Also, in line 54, I did not understand why use the order present in fruit flies, that is a derivate group (Drosophila yakuba), as the example of the most spread gene order in insect mtDNAs. There are many works mentioning the ancestral insect gene order (pancrustacean) - Cameron et al 2006; Cook et al 2005 (DOI:10.1098/rspb.2004.3042), for example.

Experimental design

The research is within the scope of the journal and the methods are sufficient described. About the research question, one of the motivation to sequence the S. longialata genome was: “the other three published genomes show substantial gene rearrangements and it is unclear if the mitogenome of any of these species represents the basal condition in the Ascalaphidae”. What are these substantial rearrangements? Because the rearrangements cited in the references were the same found to other Neuroptera species. So, related to the morphology of the genome, all the species in this family are equal.

Validity of the findings

In the Results/Discussion of the Phylogenetic analysis, specifically in the lines 250 to 252, is not clear what was the previously controversial relationship between/within Myrmeleontidae that was solved in the present work. This is one of the most important results, so it could be more emphasized, better explained in the discussion or included in the background what was the issue. And it could be included a comparison with the other works that presented the controversial results.

Also, it is not clear why the increase in number of species included in the presented analysis resulted in different topologies related to Osmylidae, Sisyridae and Nevrorthidae found in Wang et al 2017 (lines 256-258). Mainly because the species included here were from the families: Myrmeleontidae, Chrysopidae and Ascalaphidae. Also, Wang et al 2017 mentioned the use of the heterogenous model (CAT-GTR) in the Bayesian inference as the only method able to recover Neuroptera as a monophyletic group. The differences found in this work would be due the different model used and not because of the species included here.

---

## Round 0.2 · Minor Revisions

I thank the authors for their improved submission. The two reviewers found that the manuscript is much improved, however, they still point out the need for a number of corrections and clarifications. I am confident that the authors will be able to implement these constructive suggestions succesfully.

Best regards,

Reviewer 1 ·

Basic reporting

Please, refer to general comments to the author.

Experimental design

Please, refer to general comments to the author.

Validity of the findings

Please, refer to general comments to the author.

Additional comments

In my opinion, the authors have done a nice job. I believe that the authors have addressed my comments and suggestions appropriately. However, I have some minor suggestions below that I believe can improve the manuscript.

1. MATERIALS AND METHODS
1.1. Sample origin and DNA extraction:
*Line 87: You can remove the “an” before the name of the DNA extraction kit
*Line 109: I think the citation of MITOS is still not correct. The citation of the software should be after the name of the software. MITOS (Bernt et al., 2013). If you want to cite the webserver page, you can write MITOS (Bernt et al., 2013, available at http://mitos.bioinf.uni-leipzig.de/index.py).

2. RESULTS AND DISCUSSION
2.1. Mitogenome organization and structure:
*Line 149: change the word “commom” to “common”

2.2. Phylogenetic analyses: This section has been greatly improved. In the meantime, however, I have a some suggestions for minor clarifications.
*Line 253 to 254: I suggest you change “According to the opinion of Bergsten (2005)” to “According to Bergsten (2005)”. Remove the HYPERLINK from the text.
*Line 259: Please, be careful about using the term “basal” or “primitive”. I suggest removing the sentence or just saying that (Apochrysa matsumurae + Nothochrysa sp.) is a sister clade of (Nothancyla verreauxi + (Abachrysa eureka + (Chrysopa pallens + (Chrysoperla nipponensis + Chrysoperla externa)))).
*Line 263 to 264: I think it is best to change “supported” to “recovered” in this case. Another point: You did not fail to recover the monophyly. You got a result! Change “failed” to “did not recovered the monophyly of Ascalaphidae”.
*Line 267: Change “long-length attraction” to “long branch attraction”.
*Line 271: Again, be careful about using the term “basal” or “primitive”. Did you want to say that Osmylidae is sister clade of all other Neuroptera?
*Line 273: Change “supported” to “recovered”.
*Line 276: It's not clear what you mean by the phrase “were united with”. Do you want to say that they are sister clades? If yes, you need to make it clearer.
*Line 278 to 279: It's not clear what you want to say in this sentence. Do you mean that your results corroborated the results obtained by Wang et al. (2017) and Song et al. (2018)? If so, you need to make it clearer. However, it is important to note, your results did not recover the sister relationship of Myrmeleontidae and Ascalaphidae. Actually, Ascalaphidae is not monophyletic.
*Line 280 to 283: From the results and discussion of the manuscript, it does not seem that the relationship between/within Myrmeleontidae and Ascalaphidae is resolved. Futhermore, a different and/or higher sampling, as well the use of different data sources, may generate incongruent results. I suggest you reformulate this sentence to improve the discussion.
*Line 293 to 294: Again, you did not fail to recover the monophyly. You got a result! Change “failed” to “did not recovered the monophyly of Ascalaphidae”. Also, change “supported” to “recovered”. It is not clear what was also recovered by Wang et al. (2017) and Song et al. (2018). Did Wang et al. (2017) and Song et al. (2018) recovered the monophyly of Myrmeleontidae or did they not recover the monophyly of Ascalaphidae?
*Line 294: You can remove the phrase “Myrmeleontidae is inside Ascalaphidae in our results”. It is completely clear by the figures and by saying that Ascalaphidae is not monophyletic

3. CONCLUSION
*Line 302 to 304: I suggest you add the sentence that “you believe” that the cause of the differences is by the long branch attraction. I suggest this because at this moment, with the available data, the possibility of long branch attraction exists, but it is difficult to say with certainty. So, the phrase would be “From the results obtained in the present study, we believe the different topologies of phylogenetic relationships were caused mainly by long branch attraction of Coniopteryx sp., Dilar sp. and Semidalis aleyrodiformis”.

4. TABLES
*Table 1: It should be the table of Base composition of the mitochondrial genomes. However, it is presenting the species used in the work, which is the same table in the supplementary material (Table S2). Please, correct table 1.

Reviewer 2 ·

Basic reporting

The authors made all the solicited changes and solved the important aspects that were not clear concerning the motivation of the study and the evolutionary questions of the group.
Much information was added to improve the manuscript, so I would like to point out some formatting details and some parts that are not totally clear for me yet:

Figures, tables and references
Table 1 is erroneously attached. Table 1 and Table S2 are the same.

Fig 4: Add in the caption (left) and (right) to ML and BI figures.
Change “The data is includes” to “The data include”.
Also, in the text is mentioned in the lines 246 and 284 the term BI 1, I believe it is the BI from figure 4, include this in the caption.

Fig 5: Add in the caption (left) and (right) to ML and BI figures.

Review references:
Ronquist et al. 2012 is not complete in the references.
Zhang et al. 2008: include a and b in the references and in the citation.

Lines 70 – 71: “These three published genomes (Ascalaphidae) show substantial gene rearrangements (Beckenbach et al., 2008; Negrisolo et al., 2011; Cheng et al., 2014)”.
How these 3 genomes show substantial gene rearrangements if they share the same arrangement as the other families of Neuroptera? (as mentioned in Lines 148 – 156)

Experimental design

Line 121: “For the first analysis that indicated Megaloptera as a sister clade to Neuroptera, as proposed by Engel et al. (2018) and Peters et al. (2014)”.
It seems a result and that Megaloptera as a sister clade of Neuroptera is a result only from the first approach.

Validity of the findings

Lines 241-246: the low support (ML29) found between Hemerobiidae and (Berothidae+Mantispidae) is being directly compared to the high support between Hemerobiidae and Chrysopidae. So, there are two different things being directly compared.
I would suggest, in the Fig. 4, to extend the red box until Matispidae (which includes the ML29) and let clear in the text the two results: 1- The change of the Hemerobiidae’s sister clade; 2- the high support found for both relations (Hemerobiidae+Chrysopidae) and ((Hemerobiidae+Chrysopidae) + (Berothidae+Mantispidae)) only in the BI analysis.

Lines 256 – 257: “In this situation, both the ML and BI phylogenetic trees showed identical topologies and high support values for most clades, except for the internal relations within the family Chrysopidae”. The outgroup is also different.

Lines 257-261: It is important to mention that in the ML (Apochrysa matsumurae + Nothochrysa sp.) is sister clade of (Nothochrysa sp. + (Nothancyla verreauxi + (Abachrysa eureka + (Chrysopa pallens + (Chrysoperla nipponensis + Chrysoperla externa))))) whereas in BI they are all part of the same clade.
The support is low to say the clade of (Apochrysa matsumurae + Nothochrysa sp.) is the base clade of Chrysopidae. Also, in the text is not included Apochrysa matsumurae for the recovered BI analysis.

Lines 261 – 264: I would suggest change: “On the whole, this analysis highly supports the monophyly of Osmylidae, Sisyridae, Nevrorthidae, Berothidae, Mantispidae, Hemerobiidae, Chrysopidae, Psychopsidae, Nymphidae and Nemopteridae. But the monophyly of Ascalaphidae which was supported by Wang et al. (2017) failed to be supported in this study.” to something like: On the whole, this analysis supports the monophyly of all Neuroptera families except the Ascalaphidae, previously reported as monophyletic by wang et al 2017.

Lines 267-275:
Personally, I don’t think it is necessary include this deep discussion: “In the ML analysis long-length attraction existed with all families of Neuroptera (Fig. 4) and Coniopterygidae is recovered as sister clade to the remaining extant Neuroptera, which is consistent with the conclusions of Wang et al. (2017) and Winterton et al. (2010; 2018). By contrast, in the BI analysis (Fig. 4) Osmylidae is recovered as sister clade to (Coniopterygidae + (Sisyridae + Nevrorthidae). In the results of Haring and Aspӧck (2004) and Song et al. (2018), Osmylidae as the basal position of Neuroptera was supported whereas in the results of Wang et al. (2017), the relationship of (Osmylidae + (Sisyridae + Nevrorthidae)) is supported by ML and BI analyses with the homogenous GTR+I+G model. But when Wang et al. (2017) used the heterogenous CAT-GTR model in BI analysis, (Sisyridae + Nevrorthidae) + (Osmylidae + other Neuroptera) were recovered” for two reasons:
1- It is related to Fig4 and the final trees you are considering are from Fig 5.
2- These families of Neuroptera are not the focus of the study. The further discussion is more important and the reader can get lost in this part.
If this discussion will be maintained I would suggest changing it to before you start discuss fig 5 and try to summarize it.

Lines 291-294: “the monophyly of Myrmeleontidae was recovered again and is supported by Wang and Song”, but the monophyly of Myrmeleontidae is not supported by both (based on the introduction).
I would suggest to change this sentence and to make a conclusion based on the problem pointed out in the introduction (“Ascalaphidae as a sister clade of Myrmeleontidae is supported by Song et al. (2018), while Ascalaphidae within the clade of Myrmeleontidae is recovered by Wang et al. (2017)”.

Additional comments

Line 46: extra dot before the citation

Line 128: extra comma after Neuroptera

Line 133: correct the name of the program to PartitionFinder

Line 136: it lacks the citation for MrBayes 3.2

Line 156: correct 218 to 2018

Lines 164-165: “The greater length of the S. longialata mitogenome is due largely to 16 intergenic gaps ranging”
I would suggest change “gaps” to intergenic regions as you used in other topics.

Lines 220-221: “whereas the tRNA genes of Ascaloptynx appendiculatus almost displayed the typical cloverleaf secondary structure.”
What does it mean “almost” displayed the typical cloverleaf secondary structure?

Line 253 – 253: unformatted citation (“hyperlink…”)

Line 256: Change Figs. 5 to Fig 5.

---

## Round 0.3 · accepted · Accept

Thank you for the final corrections. The manuscript is now suitable for publication.

#